# MODEL COMPARISON OF BEER DATA CLASSIFICATION USING AN ELECTRONIC NOSE

## ABSTRACT

Olfaction has been and still is an area which is challenging to the research community. Like other senses of the body, there has been a push to replicate the sense of smell to aid in identifying odorous compounds in the form of an electronic nose. At IBM, our team (Cogniscent) has designed a modular sensor board platform based on the artificial olfaction concept we called EVA (Electronic Volatile Analyzer). EVA is an IoT electronic nose device that aims to reproduce olfaction in living begins by integrating an array of partially specific and uniquely selective smell recognition sensors which are directly exposed to the target chemical analyte or the environment. We are exploring a new technique called temperature controlled oscillation, which gives us virtual array of sensors to represent our signals/ fingerprint. In our study, we run experiments on identifying different types of beers using EVA. In order to successfully carry this classification task, the entire process starting from preparation of samples, having a consistent protocol of data collection in place all the way to providing the data to be analyzed and input to a machine learning model is very important. On this paper we will discuss the process of sniffing volatile organic compounds from liquid beer samples and successfully classifying different brands of beers as a pilot test. We researched on different machine learning models in order to get the best classification accuracy for our Beer samples. The best classification accuracy is achieved by using a multi-level perceptron (MLP) artificial neural network (ANN) model, classification of three different brands of beers after splitting one-week data to a training and testing set yielded an accuracy of 97.334. While using separate weeks of data for training and testing set the model yielded an accuracy of 67.812, this is because of drift playing a role in the overall classification process. Using Random forest, the classification accuracy achieved by the model is 0.923. And Decision Tree achieved 0.911.

## 1 INTRODUCTION

From the five major human senses, olfaction is the most complex sense to be understood completely. Olfaction in an electronic nose has been studied since the 1960s and the term electronic nose was first used in 1988 [1]. An electronic nose is a device that mimics the olfaction system in living beings, particularly humans by integrating chemical smell array of sensors and a machine learning algorithm to learn the patterns/ fingerprints of an odor and correctly classify the respected odor as such [2]. Initially the approach that was embraced in the gas sensing field was a one sensor for one odor type approach, as illustrated by the narrow gaussians on Figure 1. Using this approach, each sensor is very specific to a given odor. However, this was found to be a limiting approach, and it is also very challenging to fabricate highly selective sensors [3].

A much more powerful approach, is to rely on partially specific sensors, as illustrated by the graph on Figure 2 [4]. Each sensor has some specificity, but the sensors do overlap. In this case, whats important is the aggregate responses of each of the sensors which creates a unique fingerprint when combined. The number of sensors in the array depends on the application targeted.

There is a large range of applications where an electronic nose can be beneficial, simply because gases surround us all the times, and they are involved in so many processes [5]. Just to name a few general use cases, we can think of food  agricultural industry where an electronic nose could help

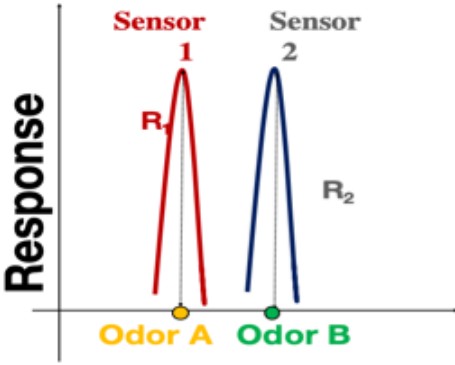

Figure 1: Narrow gaussians representing one sensor for one odor type approach

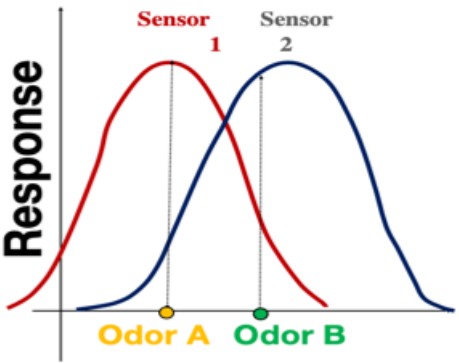

Figure 2: Wide gaussians representing overlap between sensors

detect food spoilage, and limit food wastage, crypto anchor and block chain, air quality monitoring in industrial or automotive spaces, process monitoring and standardization in the pharmaceutical or chemical industries, and healthcare, where enabling the early detection of diseases could save many lives.

In this paper, we discuss in detail an electronic nose device we called Electronic Volatile Analyzer (EVA). EVA is currently in the research phase being developed under a team called Cogniscent in IBM Almaden Research Center. On this study, we used EVA to study and classify different brands of beers based on their volatiles they emit.

## 2 SYSTEM DESCRIPTION

The proposed electronic nose, Electronic Volatile Analyzer (EVA) was developed by IBM research team. In IBM research, the team researches on three very important areas to enhance the devel-

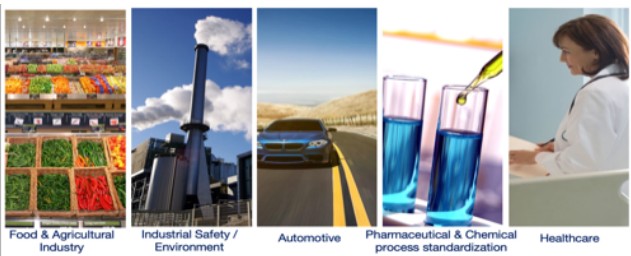

Figure 3: Impact of gas detection technology on different applications

opment of an electronic nose. Material, hardware and analytics (machine learning or Artificial Intelligence (AI)).

EVA is designed in house to fit the specification of an electronic nose which is modular, tunable, portable and scalable with low cost electronic components. EVAs modular architecture allows for the integration of more sensors depending on the application targeted. A sensor is mounted on its own module and operated in novel ways to extract relevant information and optimize sensitivity and selectivity. Each sensor is operated in a temperature oscillated wave form by applying a voltage to the heater to form one sequence. And multiple examples are collected in such a way for training. The BeagleBone serves as a central hub device, where the processing and integration of all the data from the array of sensors is happening and it communicates with a I2C protocol with the sensor modules. The platform is agnostic of sensors technology, which means we can add other types of sensors such as temperature, humidity sensors to enhance the AI intelligence and aid with more information. The system is fully integrated and independent. EVA communicates with the cloud using a Wi-Fi connection and this provides various advantages such as centralized processing, online storage and supporting complex algorithms.

The data collected by EVA is preprocessed and analyzed to extract relevant information that helps in the differentiation of the odors being detected to create a finger print. Once the features are extracted, the next step is the machine learning algorithms, the information is fed into a machine learning model for training which takes days to collect good data with enough number of examples to train the model. Once the model is trained, whenever EVA is exposed to a sample that its trained on but a fresh sample EVA never was exposed to initially, EVA will respond with the correct label for that smell.

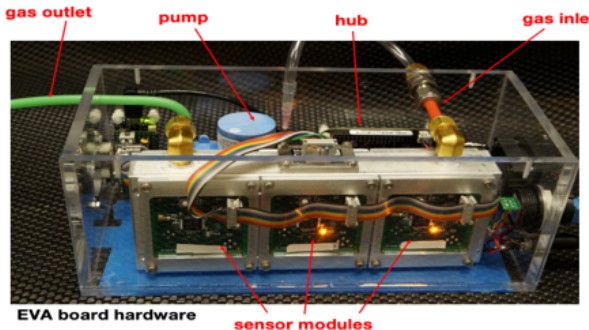

Figure 4: Electronic Volatile Analyzer (EVA) e-nose device

## 3 METHODS

### 3.1 DATA ACQUISITION

When collecting data, protocol of data collection is very important as it determines the reproducibility of the training data to have a good model for training the artificial network. The setup we use for

data collection is automated to collect multiple examples of data in a controlled mechanism. The setup has an auto-vial selector, to select vials based on the time and sequence of data collection. Some protocol we follow and make sure that they are always at the same level before we start measuring the required measurements are the temperature of our samples, the flow of the system, the gap between the syringe and the sample etc. In such an automated way we collect multiple examples of our samples over night or throughout the day for multiple days and we prepare the data for training.

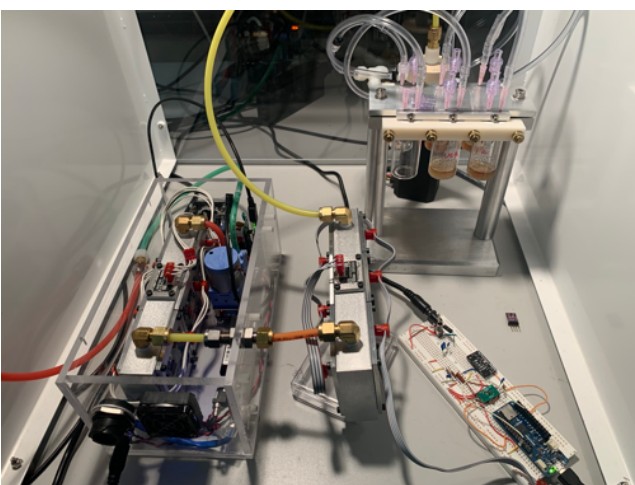

Figure 5: Setup of data collection with Electronic Volatile Analyzer (EVA) e-nose device

Data is collected from three different brands of beers, Urquell which is a pale lager with an alcohol content of 4.4 percent, Lagunitas is a craft brewery with an alcohol content of 6.2 percent and Race is a hoppy IPA brewed with malted barley, wheat and crystal malts with an alcohol content of 7.5 percent. All samples are poured into a vial and only the head-space is measured as seen from figure 5. And we collect data from lab air from a separate vial as a baseline.

### 3.2 DATA PROCESSING

After the data is collected using the setup and method described above, the data is ready for preprocessing before being fed to a machine learning model. This is one of the stages where most of the time is spent, because extracting relevant information and features that help for the differentiation and classification of different samples is very important to learn patterns for the machine learning model. We extract different types of features from our raw input data. The way we process our data is as follows: we separate the transitions from the sequences for all the sensors and we chunk the transition, which is a voltage drop or gain from two different voltage values.

And we extract a mean value and a slope value from each chunk. We chunked a transition in to five ways for this study and we extracted two features from each chunk, therefor we get ten features from every transition. The total number of features depend on the aggregate number of transitions from each sensor. Finally, after stitching all the features from all sensors, all the samples are labeled according to their ground truth labels and all the examples are stitched together to have a final training set which is the total number of examples by the number of features plus the labels.

### 3.3 MACHINE LEARNING RESULTS

On this study we used multi-level perceptron (MLP) artificial neural network (ANN) model. We had our input layer, two hidden layers and an output layer. For the first experiment, we collected data for one full week, and we split the data into 70 percent training set and 30 percent testing set, to train the model with the training data. The model further splits the data in to 85 percent actual training set and 15 percent testing set. The model took 198 epochs to finish training and it achieved a 100

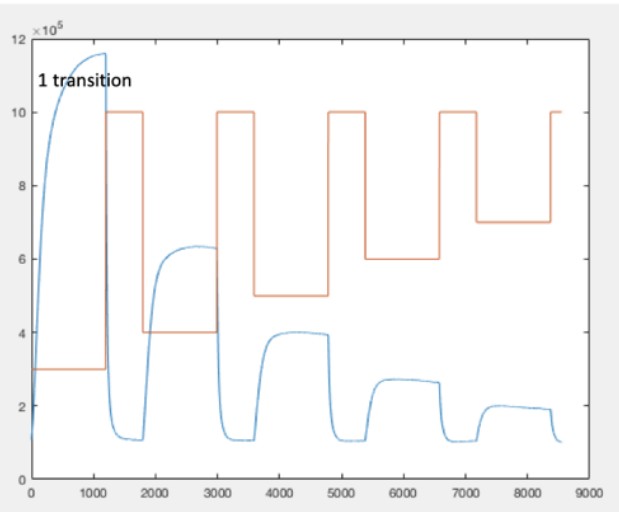

Figure 6: An example of a temperature oscillated sensor read-out from a sensor

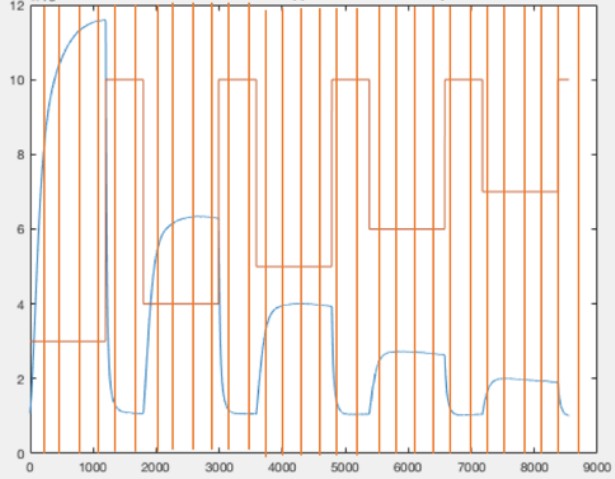

Figure 7: An example of a temperature oscillated sensor read-out from a sensor chunked to 5 equal parts per transition, which will be 30 chunks for the whole sequence from this sensor

percent accuracy with a learning rate of 0.25 and a regularization parameter and mini-batch size of 1 (see figure 8).

After training we further tested the model with the remaining 30 percent data, we kept from the data collected, which is the final days of data from that week and the model achieved a 97.334 percent accuracy by correctly classifying the different brands of beers plus lab air.

For the second experiment we kept the conditions the same, but we continued collecting data for another week and we used the first week data as a training set and the second week data as a testing set and it yielded 67.812 percent. For this specific case we see the accuracy dropping and based on our research studies the reason is because of drift. And this is because of multiple factors playing a role, such as sample changing through time, sensors not starting from the same baseline etc.

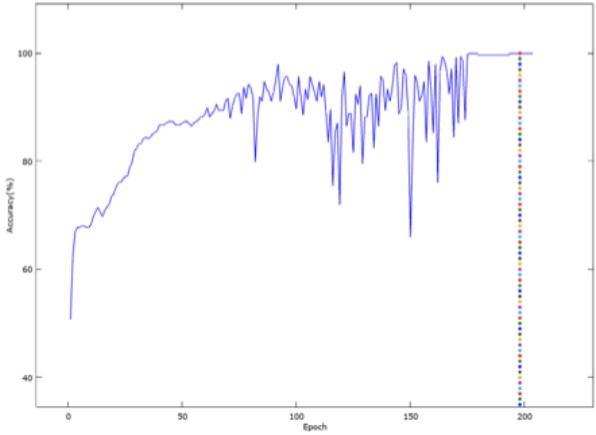

Figure 8: Epoch accuracy test Beer one-week data

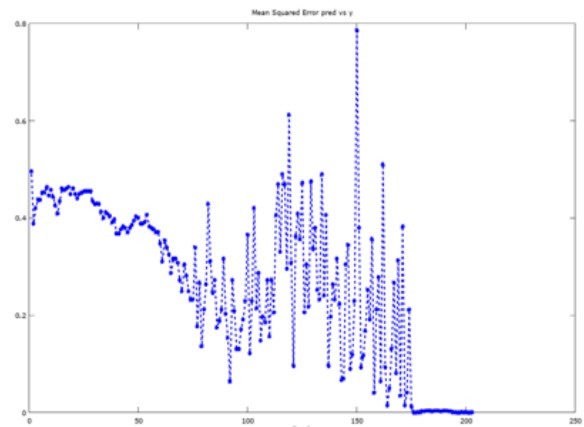

Figure 9: Mean squared error prediction vs ground truth

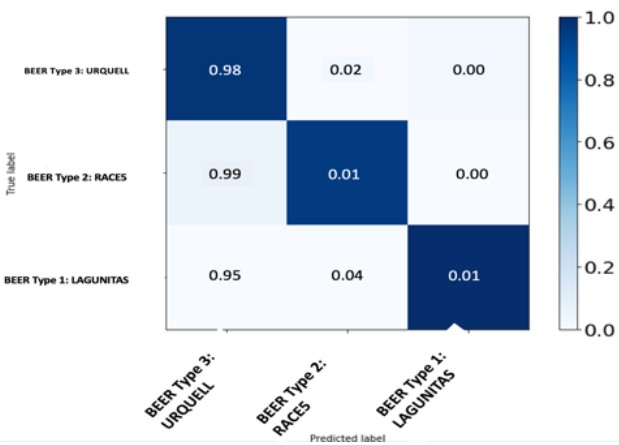

Figure 10: Confusion matrix of different brands of beer plus lab air one-week data

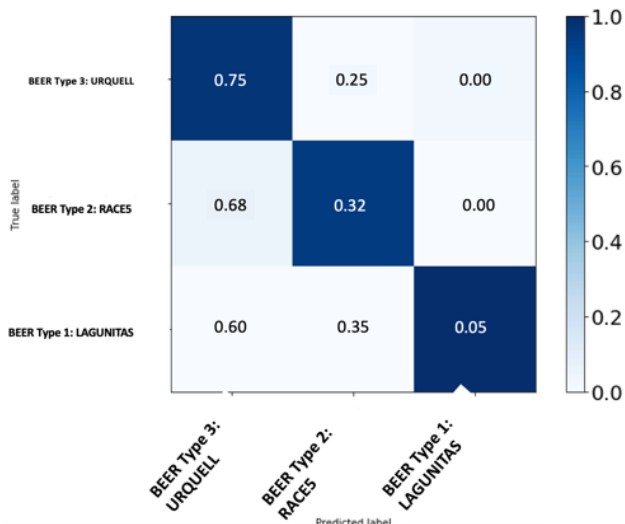

Figure 11: Confusion matrix of different brands of beer plus lab air two-weeks data

## 4 CONCLUSION

In this study, a portable electronic nose device is introduced and the device we called EVA is used to classify between different brands of beers. And as seen from the experiment results using an electronic nose for identification of different substances based on their smell is very promising.

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
