# OpenReview forum: "Model Comparison of Beer data classification using an electronic nose"
_ICLR.cc/2020/Conference — Reject_

### Official Review · AnonReviewer3 · 2019-10-18
**Official Blind Review #1376**

**Rating:** 1

**Review:**

This work designs an electronic nose to classify beers and gets extremely high accuracy through neural network methods.
However, it seems that ICLR is not a proper conference for such a work.
The main contribution of this work may be the electronic devices rather than the machine learning methods.
The author related the results of random forest but there is no related content in the main body of the paper.

At last, the paper is short of technical contents and the figures occupy a lot of the space.

Overall, I am afraid that this work should find a better publication site other than ICLR.


**Experience Assessment:**

I do not know much about this area.

**Review Assessment: Checking Correctness Of Derivations And Theory:**

I assessed the sensibility of the derivations and theory.

**Review Assessment: Checking Correctness Of Experiments:**

I carefully checked the experiments.

**Review Assessment: Thoroughness In Paper Reading:**

I read the paper at least twice and used my best judgement in assessing the paper.

---

### Official Review · AnonReviewer2 · 2019-10-20
**Official Blind Review #2**

**Rating:** 1

**Review:**

Main comments:
Quality: This paper reads like a project report rather than a typical ICLR paper. It describes a direct application of naïve learning tool (random forest and neural network) without sufficient insight.

Clarity: In general, the descriptions on data collection and training are clear.

Originality: The novelty of this work is very limited, if any.

Contributions:
The contribution are limited. If the main contributions lie it the construction of the device, I suggest a resubmission to a device-related conference.

From the machine learning perspective, this work is a simple and direct application of existing machine learning tools..


Other comments:
This paper mentions the authors are from a group (Cogniscent) at IBM, which violates the double-blind policy (at least to certain degree).


**Experience Assessment:**

I have published in this field for several years.

**Review Assessment: Checking Correctness Of Derivations And Theory:**

I assessed the sensibility of the derivations and theory.

**Review Assessment: Checking Correctness Of Experiments:**

I assessed the sensibility of the experiments.

**Review Assessment: Thoroughness In Paper Reading:**

I read the paper at least twice and used my best judgement in assessing the paper.

---

### Official Review · AnonReviewer1 · 2019-10-29
**Official Blind Review #1**

**Rating:** 1

**Review:**

This paper violates the conference's double-blind reviewing policy by explicitly identifying their research institute and team name. The violations occur in the 4th line of the abstract, the last paragraph of Section 1 (Introduction), and the first paragraph of Section 2 (System Description). For this reason, I am not providing a review of this paper.

.........................................................................................................................................................................................................

**Experience Assessment:**

I do not know much about this area.

**Review Assessment: Checking Correctness Of Derivations And Theory:**

N/A

**Review Assessment: Checking Correctness Of Experiments:**

N/A

**Review Assessment: Thoroughness In Paper Reading:**

N/A

---

### Decision · Program_Chairs · 2019-12-19

**Decision:**

Reject

**Comment:**

The paper has received all negative scores. Furthermore, one of the reviewers identified an anonymity violation. This is a reject.